# Understanding the Predictors of Economic Politics on Elite Sport: A Case Study from Spain

**DOI:** 10.3390/ijerph191912401

**Published:** 2022-09-29

**Authors:** Jordi Seguí-Urbaneja, David Cabello-Manrique, Juan Carlos Guevara-Pérez, Esther Puga-González

**Affiliations:** 1National Institute of Physical Education of Catalonia (INEFC), University of Lleida (UdL), 25192 Lleida, Spain; 2Grup d’Investigació Social i Educativa de l’Activitat Física i de l’Esport (GISEAFE), INEFC, University of Barcelona (UB), 08038 Barcelona, Spain; 3Department of Physical Education and Sports, Faculty of Sport Science, University of Granada, 18011 Granada, Spain; 4Faculty of Economics and Business, University of Zaragoza, 50005 Zaragoza, Spain; 5IGOID Research Group, Department of Physical Activity and Sport Sciences, University of Castilla-La Mancha, 45071 Toledo, Spain

**Keywords:** sport federations, sport system, organisational performance, finance, active life

## Abstract

Sport federations (NSFs) are the main promoters of sport at a national level. Their complex management involves coordinating relations with private entities, public administrations, and international organisations. Therefore, the economic situation of a country and its sport support policies have significant influences on the achievement of the NSFs’ objectives and, therefore, on their sustainability and influence on an active population. This study analyses the determinants of the financial performance of 59 Spanish sport federations (SSFs), 28 Olympic and 31 non-Olympic, based on the relationship between the funding received and their international results during the period from 2007 to 2019 (both years included). The preliminary data analysis included an examination of the missing data, and a *t*-test was used to compare Olympic and non-Olympic sport federations regarding different variables related to their resources and results. In addition, multiple linear regressions identified the possible predictors of the financing of sport federations and were separately performed for Olympic and non-Olympic federations. The results showed that SSFs were able to maintain their results in the face of decreasing resources. In addition, Olympic SSFs were found to be less dependent on public funding than non-Olympic SSFs for competitive results. This is evidence of a paradigm shift in the management of Spanish federated sports, evolving towards a model that is less dependent on the state, more efficient, and therefore more sustainable.

## 1. Introduction, State of the Art and Literature

The 1992 Barcelona Olympic Games marked the beginning of a revolution in the world of sports on an international level: the transition from amateurism to professionalism [1]. In recent decades, elite sports have been involved in a context of organisational changes based on improving effectiveness, efficiency, and competitiveness, where the adoption of private sector management techniques is very common [2]. All this has been and is a consequence of the pressure from the competition, sponsors, the media, the consumer and competitive society, and governments implementing public policies on elite sports.

Elite sports are managed through national sport federations (NSFs), each with its own sporting modality, by subrogating in them the public policies, and maintaining, in most cases, total economic dependence on public subsidies [3]. Over time, the shortcomings of the sector [4] have contributed to the consolidation of a widespread management model characterised by coercive pressure from governments through the funding of NSFs [5,6,7]. In this sense, the influence of the environment will depend on each specific national sport system, as not all states have the same system. The literature makes a clear distinction between two systems: (a) state-organised business systems, where the state plays a more interventionist role in structuring economic activities, characteristic of continental European countries; and (b) liberal market economies, characterised by the presence of a capital market-based financial system, the absence of burdensome regulations and high-trust relationships, characteristic of the Anglo-Saxon world [8].

This is a key aspect when contextualising the present study, since being subscribed to the Spanish environment—a bureaucratic system—we must consider the differences with respect to other countries in the Anglo-Saxon environment, i.e., entrepreneurial systems [9]. These differences have been highlighted in previous studies that contrast French and English [10] or Spanish and English systems [9]. All these studies confirm that most National Governing Bodies (NGBs) in continental Europe are federations run as public services delegated by the state and are, therefore, controlled by the government through the Ministry of Sport [7]; whereas UK NGBs are independent of the government [11,12]. Following this taxonomy, the continental European sport system is classified as “bureaucratic” because of the very active role played by governments in regulating the system; by contrast, the UK sport system is considered “entrepreneurial” because the role of governments is limited to establishing a framework that allows the logic of the market to express itself [10].

### 1.1. Elite Sport

In the literature, the terms elite sport, federated sport, top-level sport, or high-performance sport are used interchangeably as synonyms. For the purposes of this study, the term elite sport is used, which is understood to mean a competition involving athletes who receive subsidies and/or public support so that they can devote themselves entirely to training and representing the respective state in their competitions [13].

It is increasingly evident that elite sport has gone global and has become an indicator of the potential of countries, where sporting results are interpreted as the economic, social, and cultural power of a new geopolitical organisation [14]. Some authors argue that sport should become an active instrument of institutional communication, as a country brand [15], the clearest example being the UK’s sporting model, assimilated by other advanced states, called the “virtuous cycle” of sport, whereby elite sporting success leads to both the international prestige of the nation, a “feel-good factor” among the population and, most importantly, an increase in mass participation and an active life [16].

The competition between nations to improve their sporting performance in the international context has led to a race to find out what factors determine success in the elite sport system [17,18]. It should be noted that experts have come to accept that there is no single, ideal sporting model. Under this premise, previous studies have focused on the degree of importance that different factors and their associated variables have on the achievement of success [19,20]. According to Amis [21], there are six main categories of resources from which sustainable competitive advantage can be derived: (i) financial; (ii) physical; (iii) human; (iv) technological; (v) organisational; and (vi) reputational. Swann et al. [22], for their part, proposed a framework to evaluate elite sports. Their framework includes five variables (competitive standards, competitive success, experience, the competitiveness of a sport in the athlete’s country, and the global competitiveness of that sport), each allocated a score, with an accumulated score classifying eliteness. Inevitably, the variables and weightings in the proposed framework are arguable, but it is a strong contribution to providing a consistent definition of elite status for further development (as the authors recommend).

In addition to the studies focused on identifying the factors that determine the success of elite sports, comparative studies between different countries have been carried out. The conclusions of these studies coincide in considering funding as a determining element and one that is clearly linked to the success of the system [23,24,25]. In addition, some authors have studied the relationship between the efficiency of funding and sporting success [26,27]. From the published research, it appears that nations are increasingly investing in a planned way by applying a systematic approach to elite sport policy [28]. In this regard, as is evident in Figure 1, although the number of club licences has declined, the production of high-elite athletes (DANs) in relationship with federation licences (registered players) has considerably increased in recent years.

In this context, European national sport systems, conceived with the aim of promoting healthy physical activity to increase well-being among the more active inhabitants, have, after the eruption of neoliberal economies, become the means to implement public policies aimed at supporting private initiatives to promote high performance and disseminate sport practice [29].

This situation has resulted in NSFs having to debate the dilemma of managing elite sports or mass participation sports, where the focus of the former is on “sport development”, while the latter is on “sport for development”, the two policies being incompatible and the common management of them non-peaceful [30].

### 1.2. Elite Sports in Spain

In Spain, elite sports are organised through public and private structures, the governing bodies of which are, respectively, the Spanish Sport Council (Consejo Superior de Deportes-CSD) and the Spanish Sport Federations (Federaciones Deportivas Españolas; SSFs). The SSFs, legally speaking, are private, non-profit associative entities with their own legal personality and assets, independent of those of their members [31], to which public administrative functions are attributed, acting in this case as collaborating agents of the public administration [32]. One of the public functions of an administrative nature delegated by the state to the SSFs is the organisation and development of elite sports [31]. To carry out their functions, the SSFs have two sources of financing: (a) public funds through subsidies from the CSD and other additional channels, such as the Olympic Sports Association (ADO) and the ADO aid for Paralympic Sport (ADOP) programmes, to be considered by the state for the exercise of public functions; and (b) other private additional sources and those from international organisations; both of these sources make up their financial structure.

The Spanish government’s financial aid distribution policies are diffuse and non-fully transparent. In its public calls for proposals, the CSD establishes a series of criteria related to two main areas for obtaining financial aid: (a) the level of sporting results, and (b) the level of development and management of sport. However, the objective indicators that allow both variables to be assessed are not published, which in fact turns the public aid system into a politicised system in which the government in power can give and distribute as it sees fit [33].

The system institutionalised in Spain in 1990, with the publication of Law 10/1990 on sport, has meant that most SSFs are highly dependent on public subsidies. Thus, in 2012, more than 90% of the SSFs were in a situation of technical bankruptcy, as they had negative working capital and negative net worth [34]. In 2013, this situation led the CSD to implement measures to guarantee the economic viability of the system, thus obliging those SSFs that were in that financial situation in 2012 to develop a viability plan, failure to comply with which would result in a penalty on the amounts of CSD subsidies during the years the plan was in force [7]. The consequence of the application of financial viability programmes by the SSFs was the accountability to the CSD and the essential measurement of sporting performance. This situation forced the SSFs to move from a purely administrative function—receiving and distributing public subsidies—to being accountable for the purpose of the public subsidies, i.e., managing sports and organisational performance. They went from using public resources for their own interests to managing public resources in the interest of the state [35].

The failure of SSFs to comply with viability plans led to a reduction in public subsidies from the CSD [36], and the reduction in available financial resources helped to drive the process of the professionalisation of the sector at the national level [2]. SSFs adopted structures capable of maintaining, in an environment of austerity, the same competitive results, showing improvements in their performance both in terms of effectiveness and efficiency [3]. This qualitative leap in the performance of the SSFs over the past decade has been key to the sustainability of elite sports in Spain.

In the organisational context, performance is the combination of effectiveness, as the organisation’s capacity to achieve its objectives, and efficiency, as the ratio between the resources used and the results obtained by the organisation [37,38]. In addition, organisational performance is recognised by theorists and practitioners as a multidimensional concept framed by a set of simultaneous contradictions that, in the case of sports and for NSFs, are presented as tensions between paid staff and volunteers, elite and mass sports, public and private funding, and social and commercial cultures [30]. In fact, the achievement of institutional objectives is often more complex for NSFs than for other non-profit organisations, as it involves the integrated action and intervention of national and international governing bodies, leagues, clubs, and national, regional, and local governments [37].

### 1.3. The Economic Sustainability of Spanish Elite Sports

Sustainability requires a triple bottom line approach, according to which improvements are pursued in the social, environmental, and economic dimensions of performance, and NSFs are organisations with social and environmental goals and a growing need to manage their resources efficiently, highlighting the importance of the financial pillar as a necessary factor to achieve the traditional purposes of NSFs [7].

Some authors have defined economic sustainability as the ability of an organisation to manage its resources and generate profitability in a responsible and long-term manner [39,40]. Other authors have conducted studies to identify economic indicators for sustainability [41,42].

In elite sports, the capacity of the governing bodies, namely the Board of Directors and the General Assembly of the SSFs, to obtain and manage their financial resources is one of the crucial dimensions for achieving their strategic objectives [43]. This statement could be explained by the fact that inputs and outputs (or means and ends) are inverse in for-profit and non-profit organisations. Therefore, while profit represents the main output for for-profit organisations, non-profit organisations use financial resources as an input to their operations [17,35,38]. Therefore, the criteria with which these resources are used will be decisive for the sustainability of SSFs [44].

In this sense, the accentuated dependence of the SSFs on public subsidies to carry out their activity [6,7], and the need to obtain the maximum benefit from those resources in terms of competitive results [45], generate a scenario of maximum demand for the management of the NSFs.

Nevertheless, the periods to which this study subscribes have coexisted in the context of the global financial crisis that began in 2007 and the Spanish economic depression that began a year later, paving the way for the financial situation of an elite sport with a high dependence on public funds [7]. In order to face this context, a series of policies were required: (i) the cleaning up of the accounts of SSFs through the implementation of viability plans in 2013 for those SSFs with a loss-making financial situation [34]; (ii) the publication of this financial information in compliance with the Transparency Law 19/2013 [46]; and, (iii) the adoption by the CSD of those measures that prioritised SSFs considered to be of “national interest”, including all Olympic sports (federations) and a small universe of non-Olympic sports (federations), which aggravated the situation for the latter compared with the rest, experiencing an average reduction of 42% in their public funding between 2011 and 2013 [36].

Nevertheless, this arduous period bore fruit, replacing the classic model of public management prevailing in Spanish federated sports with a new model of public management. This new model applied private sector strategies that entailed the gradual replacement of a part of the volunteer staff by paid staff, with their consequent professionalisation; the increase in the disclosure of sensitive information to stakeholders in terms of transparency; and the move from pre-budgetary accounting to accrual accounting in terms of accountability. This effort resulted in strengthening the economic sustainability of the sector [7], from a loss of 4.8 million euros in 2012 to a surplus of 6.7 million euros in 2016 [34], and a sustained decrease in debt levels, which reached its historical maximum in 2020, as can be seen in Figure 2.

It is from this context that this study aims to analyse the determinants of the financial performance of SSFs based on the relationship between the funding received by Spanish elite sport entities and their success. In this respect, any predictor of the behaviour of SSFs is a particularly complex task, both because of the differences in the characteristics of the various federations and because of the very nature of sport activity, which does not respond to a standard productive function that links the resources consumed with the results achieved. Therefore, the study of federated sport requires a multidimensional approach, understood as “the capacity to adequately acquire and process human, financial and physical resources to achieve the objectives of the organisation” [38], for which multivariate models are very useful when carrying out studies in the sector, and this is the methodology used for the development of this study. To carry out the study, economic funding was considered as monetary aid in the different forms that the CSD has detailed, recognised, and published. Additionally, to determine sporting success, the results obtained in Olympic Games, World Championships, and European Championships were analysed.

## 2. Materials and Methods

### 2.1. Data Collection

Data were collected from 59 sport federations which were separated into Olympic (28) and non-Olympic (31) according to the nature of the elite sport they represented.

For each federation, we collected the data from 2007 to 2019, both years included, analysing their official budgets and the CSD budgets together with the ADO grants. Different target variables were included in the database representing three different blocks: (a) sporting results (i.e., quality performance and the number of medals from the previous year); (b) sporting structure (i.e., number of sport licences, number of high-level athletes, and number of clubs); and (c) financial resources (i.e., the total financing and CSD financing). Table 1 describes each of these variables.

### 2.2. Data Analysis

First, the preliminary data analysis included the examination of the missing data. Second, we obtained the descriptive statistics of each variable in the study, including the means, standard deviations, and correlations. Third, we conducted a *t*-test to compare Olympic and non-Olympic sport federations regarding the variables of the study. The Bonferroni correction was introduced to interpret the results of these comparisons.

Finally, we ran a series of multiple linear regressions to identify the possible predictors of the financing of sport federations. In these analyses, the variables related to sporting results and sporting structure were considered the potential predictors of the total budget and CSD funding of the federation. Multiple linear regression analyses were separately performed for Olympic and non-Olympic federations. All the tests of significance were two-tailed, and statistical significance was defined at *p* < 0.05 unless otherwise indicated. All the analyses were performed using the IBM SPSS statistical package, version 23 (SPSS, Chicago, IL, USA).

## 3. Results

### 3.1. Preliminary Analysis

First, we examined the missing data. Two sources of missing data were identified. On the one hand, different federations were constituted after 2007, and therefore, data could not be obtained for the years before their constitution. On the other hand, the missing data came also from the unclear or incoherent information collected from the federations, which was labelled as the missing data. Table 2 presents the descriptive results for each variable in the study and the correlations between the variables in the study. High and moderately high correlations were found between the number of medals and the quality of performance and between the number of sport licences, the number of clubs, and the total budget. Overall, it may be noted that the pattern of correlations was different for the total budget and for the CSD funding.

### 3.2. Comparison between Olympic and Non-Olympic Sport Federations

Table 3 presents the comparison between Olympic and non-Olympic sport federations regarding the variables of the present study. After applying the Bonferroni correction, significant differences were observed in the number of medals (*p* < 0.001), quality performance (*p* = 0.001), the number of high-level athletes (*p* < 0.001), and the CSD funding (*p* < 0.001).

### 3.3. Predictors of the Financing of Federations

A series of linear regressions were run to assess the direct effects of sporting results and sporting structure variables on financial variables. The analysis was completed in two steps, with the number of medals and quality performance included in Step 1, and the number of high-level athletes, the number of sport licences, and the number of clubs added in Step 2. The use of this sequence allowed us to identify the unique variance added by each set of variables in the model. Regression analyses were separately conducted for Olympic and non-Olympic sport federations.

#### 3.3.1. Olympic Sport Federations

Two multiple linear regressions were calculated to predict the financing of Olympic federations (see Table 4). A significant regression equation was found both for the total budget (F [5, 326] = 104.958, *p* < 0.001) and the CSD funding (F [5, 326] = 32.594, *p* < 0.001). The equation for the total budget returned an R2 of 0.617 and for CSD funding, an R2 of 0.333. On the one hand, the number of high-level athletes (β = 0.259) and the number of sport licences (β = 0.608) were significant predictors of the total financing received by the Olympic federations. On the other hand, the number of high-level athletes (β = 0.474) was the significant predictor of the financing received from the CSD.

#### 3.3.2. Non-Olympic Sport Federations

As the last step of our analysis, we run two multiple linear regressions to find the predictors of the financing of non-Olympic federations (see Table 5). Significant regression equations were found for the total budget (F [5, 376] = 561.791, *p* < 0.001) and the CSD funding (F [5, 376] = 25,461, *p* < 0.001). The equation for the total budget returned an R2 of 0.882 and for the CSD funding, an R2 of 0.253. On the one hand, the number of medals (β = −0.319), the number of high-level athletes (β = 0.145), and the number of clubs (β = 0.939) were significant predictors of the total funding. On the other hand, the number of medals (β = −0.794), quality performance (β = 0.933), and the number of high-level athletes (β = 0.349) were significant predictors of the funding received from the CSD.

## 4. Discussion

The data of the present study regarding the medals received in World Championships allowed us to affirm that when all the SSFs were analysed, there were no differences between the years analysed, progressively growing from 521 in 2008 and reaching 759 in 2017. However, it can be seen that the total financing (budget) progressively decreased in recent years, declining from 201 million euros in 2008 to 123 million euros in 2017, with very serious consequences, resulting in a decrease in sporting structures and projects.

These data contradict some of the studies and many of the statements that lead us to consider that the greater the financing, the greater the sporting results, demonstrating that there are other factors that affect performance, without there being a cause–effect relationship between money and results. This leads us to confirm that the level of sporting results depends on a balanced multifactorial policy in which financing is an important part but not the only determining element of sporting structure and the achievement of medals in World Championships and Olympic Games [17]. Therefore, from a causal explanation of the results of sport policies in a particular country, this study responds to the need to provide a different view of the traditional hegemonic approaches that have addressed the determinants of the elite sport policy success in the literature [48].

In this sense, SSFs have shown to be able to maintain their results in the face of a decrease in their resources, and that, in the case of Olympic federations, is reflected in the absence of a significant relationship between their sporting results (medals and quality performance) and the total funding or public subsidy from the CSD. This is commendable considering that the increasing success of countries with the same supply of medals has led to diminishing returns on investment in elite sports (mostly publicly funded) and that an increase in such investment is advisable simply to maintain the existing level of performance [49]. This result may be a consequence of the significant reduction in public aid between 2012 and 2016 mentioned by Puga et al. [36], which is logical in terms of sustainability [7], if we consider how vulnerable a policy is to the state allocating its resources to an “elite” sport that represents a small portion of the sporting universe, without a guarantee on a return on this investment in terms of any results from a sporting, economic, or social point of view [3]. This fact highlights the importance of setting limits to this support in a context in which these resources are in demand in other areas, making it necessary to evaluate the degree to which they are used [3,16]. It also explains the lower weight of public funding in the total funding of Olympic sports, with the consequent reduction in the expected relationship between the level of sporting performance and state support.

Therefore, the relationship between both the funding channels and the number of high-level athletes (DANs) can be interpreted as a strategy within the sustainability paradigm, by replacing the past “short-term” vision of investing in “ends” in terms of medals (which are uncertain), with “means”, in this case, of production within the system, since the DANs not only represent potential medal generators in competitive terms [26] but also a means of attracting private funding, mainly through sponsorships from companies that want to link their brand to a sport with an attractive competitive balance and to potentially successful athletes who allow these resources to be amortised over a longer period of time.

Similarly, the significant relationship observed between the number of licences and the total funding (β = 0.608) is subscribed to the same logic of economic sustainability, by the simple fact that “the greater the number of licences, the greater the income”, with the consequent increase in own resources, and the exploitation of the economies of scale; and social sustainability, by responding to the call for “sport for all” in terms of promoting sports to nurture the youth that constitutes the raw material of elite sports [33].

As for non-Olympic sports, their situation worsened with the economic crisis of the last decade, as they were not included in the list of sports considered to be of “national interest” [36].

It should be noted that the CSD only supports non-Olympic sports that obtain good results, with its residual support for non-Olympic SSFs that were not part of the list of “national interest”. Thus, the SSFs depended economically on their own resources through their income from the number of licences, and in the case of having DAN—meaning that there are athletes with good results—it favours the arrival of private sponsorships due to the attraction of sporting results. This situation forced the SSFs to look for ways to increase their own resources, although only a significant relationship was found between the number of clubs and the total funding (β = 0.939). However, there was a significant relationship between the CSD funding and the variables related to the level of sporting results, the number of medals (β = −0.319), quality performance (β = 0.933), and number of DANs (β = 0.145), which were clear predictors of state financial support, as only those that demonstrate a high level of competitive results will receive a subsidy, thus clearly demonstrating the relationship of dependence between public funding and results, as they are forced to dispense with long-term plans (such as those of an Olympic cycle), to attend to much shorter investment cycles that allow immediate results in terms of medals and quality of performance, at the cost of compromising their sustainability in the long term.

### Practical Implications

Taking into account the analysis carried out and the results obtained in this study, the CSD should review the public policy for the development of the Spanish elite sports system. Thus, the calls for financial aid should be linked to which objectives the SSFs should achieve: sporting results or sporting structure, or both. The public call for subsidies should be revised and should include the procedure, the specific indicators, and the algorithms for awarding public subsidies. It is certainly essential to know the process by which grants will be awarded, to help the SSFs to establish their strategic plans. Finally, the results of the grants should be publicised with the respective scores awarded to the different applications. This would ensure transparency and objectivity in the process.

On the part of the SSFs, it would be interesting that knowing the economic policy of the CSD and knowing the dependence of sporting results on economic resources, they should develop strategic plans with two priority objectives: (a) to diversify their sources of funding, they should be aware that the aid from the public administration is decreasing, and the income from licences and clubs is limited, so they should make efforts to attract private funding; and (b) to improve the sustainability of the organisation, the SSFs should continue working on the professionalisation of their staff in order to increase the efficacy of their results and the efficiency of their resources.

Considering all these dimensions, and having adequate internal and external financing, better results and greater satisfaction of all stakeholders (governments, federations, athletes, and other people, entities, and related sport organisations) can be achieved. This study, and the review of the bibliography carried out, shows that an adequate and sustained investment brings good sporting results, while an anarchic and unstable subsidy, such as the one carried out in Spain, due to the greater severity of the crisis and the lack of strategy, brings with it unpredictable and perplexing results. This, together with the lack of investments and long-term strategic policies, will mean a decapitalisation of the sporting structure and a reduction in sporting results in the short and medium terms with an indirect impact on the active lifestyle of the population who have high-performance sports as their role model for weekly practice.

## 5. Conclusions

This study analyses the determinants of the financial performance of 59 Spanish sport federations (SSFs), 28 Olympic and 31 non-Olympic, based on the relationship between the funding received and their international results during the period from 2007 to 2019 (both years included).

Taking into account the results, a reduction in the dependence of sports federations on public funds can be observed, which makes it possible to establish a more sustainable sports management model [5,6]. These changes have brought with them a possible paradigm shift, with the gradual implementation of the new postulates of public management in the context of managing a financial crisis [7], adopting private sector structures, moving from amateur to professional sport, and in many cases, replacing part of the volunteers with paid staff, with the consequent professionalisation of the sector [2].

For both types of SSFs, the conjunctures of the past decade have resulted in a maturation process, generating more efficient and sustainable structures in the face of unpredictable state funding sources. This situation persists, as these public funds continue to be undermined by the COVID-19 pandemic and the current crisis in Ukraine, forcing countries to set other priorities above sport when distributing public funds. Therefore, the economic sustainability of SSFs will depend on their ability to attract resources and obtain competitive results in terms of effectiveness [25], and the rational use of those resources in terms of efficiency [3,35,37].

Two important limitations of this research stand out. On the one hand, the study was carried out during a period (2007–2019) that can be considered short in sporting terms. That is to say, it may be that some SSFs might have obtained sporting results even though the budget was reduced because they already had the team or elite athletes in their exploitation phase and, therefore, such results were not reflected in the data. Therefore, the study should be replicated over a longer period of time, which would include the different phases of growth, exploitation, maturation, and ageing of a team or elite athlete and the relationship of these stages with funding.

On the other hand, for the data analysis, it should be considered that the details of the CSD criteria for the allocation of resources were not available, as they are not publicly available. However, considering that these criteria roughly incorporate competitive performance in both Olympic and non-Olympic sports (the more subvention, the greater the sporting results and vice versa), the results obtained in this study do not reflect this relationship. In this respect, access to these criteria will allow future studies to verify more precisely whether the grant system contributes to the achievement of the objectives expected by the NSFs [33].

## Figures and Tables

**Figure 1 ijerph-19-12401-f001:**
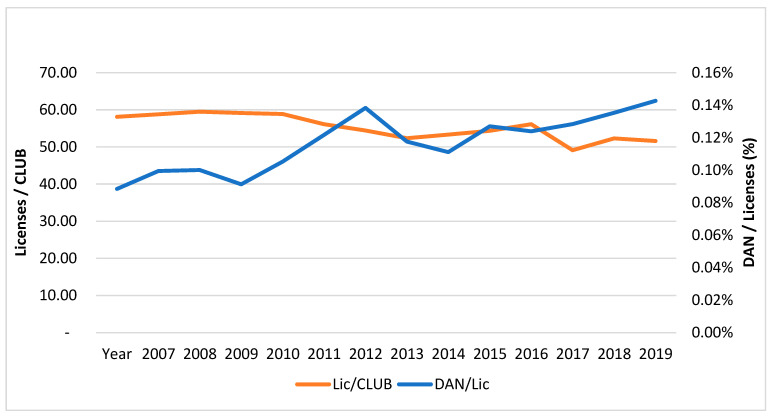
Demographic proportions of clubs, DAN and Licences over the period 2007–2020. Source: Own elaboration.

**Figure 2 ijerph-19-12401-f002:**
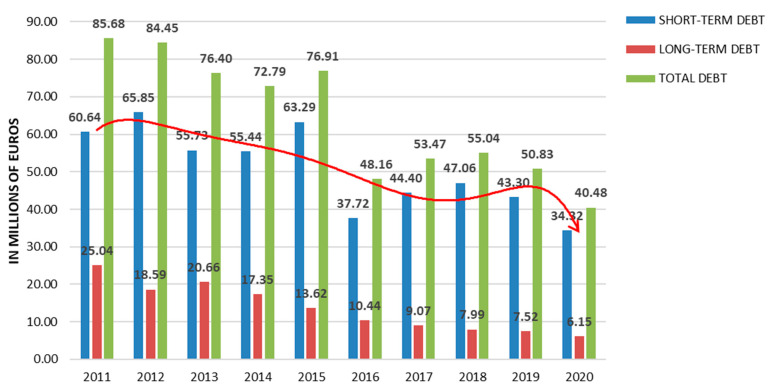
Decrease in the debt of Spanish sport federations in the period 2011–2020. Source: CSD. Subdirección General de Deporte Profesional y Control financiero (2021). The great economic figures of the Spanish NSFs 2011–2020 [47].

**Table 1 ijerph-19-12401-t001:** Variables of the study.

Variable	Type	Year	Description
*Sporting results*			
Number of medals	IV	Previous year	Total number of medals obtained in European Championships, World Cups, and Olympic Games
Quality performance	IV	Previous year	Index composed of the sum of medals obtained in European Championships, World Cups, and Olympic Games multiplied by a value that represents its relevance: 3 points for gold medals, 2 points for silver medals, and 1 point for bronze medals
*Sporting structure*			
Number of sport licences	IV	Previous year	Total number of registered players (sport licences)
Number of high-level athletes	IV	Previous year	Total number of elite athletes included in the Spanish program (high-elite athletes: *Deportista de alto nivel; DAN*)
Number of clubs	IV	Previous year	Total number of registered sport licences
*Financial resources*			
Total budget	DV	Current year	Total amount of euros managed for the federation, including own budget, financing received from the CSD, and financing received from the Plan ADO
CSD funding	DV	Current year	Total amount of euros received from the CSD

Note. IV = independent variable; DV = dependent variable.

**Table 2 ijerph-19-12401-t002:** Descriptive results and correlations between all the variables in the study.

Variable	*M* (*SD*)	Correlations					
		2	3	4	5	6	7
*Sporting results*							
Number of medals	10.4 (12.5)	0.991 **	0.523 **	0.025	−0.012	−0.013	0.338 **
Quality performance	20.0 (24.7)		0.515 **	0.037	−0.001	−0.005	0.326 **
*Sporting structure*							
Number of high-level athletes	62.8 (70.0)			0.370 **	0.317 **	0.312 **	0.583 **
Number of sport licences	56,153.6 (128,519.5)				0.854 **	0.791 **	0.138 **
Number of clubs	1097.3 (3135.1)					0.780 **	0.135 **
*Financial resources*							
Total budget (in EUR)	4,080,436.5 (14,387,529.4)						0.208 **
CSD funding (in EUR)	879,187.2 (1,046,616.9)						

Note. ** Indicates that correlation was significant at *p* < 0.01.

**Table 3 ijerph-19-12401-t003:** Comparison between Olympic and non-Olympic sport federations.

Variable	Olympic Federations *M* (*SD*)	Non-Olympic Federations *M* (*SD*)	Comparison ^1^ *t* (*df*), *p*
*Sporting results*			
Number of medals	12.3 (13.6)	8.7 (11.3)	−3.769 (646.080), <0.001
Quality performance	23.3 (26.3)	17.2 (23.0)	−3.314 (669.771), 0.001
*Sporting structure*			
Number of high-level athletes	86.9 (78.7)	42.6 (54.2)	−8.727 (578.455), <0.001
Number of sport licences	45,663.9 (71,441.2)	65,144.9 (161,399.9)	2.169 (566.494), 0.030
Number of clubs	900.5 (2413.3)	1265.9 (3632.3)	1.626 (695.401), 0.104
*Financial resources*			
Total budget (in EUR)	4,210,628.4 (4,682,041.9)	3,969,453.3 (19,109,386.9)	−0.252 (485.287), 0.801
CSD funding (in EUR)	1,532,835.9 (1,175,401.1)	321,978.5 (421,726.1)	−18.657 (442.501), <0.001

Note. ^1^ After Bonferroni’s correction, the significance level was set at *p* = 0.007.

**Table 4 ijerph-19-12401-t004:** Multiple regression model of predictors associated with the financing of Olympic sport federations.

		Total Budget					CSD Funding				
	Variables	B	95% CI for B	*β*	Adj. R^2^	R^2^	B	95% CI for B	*β*	Adj. R^2^	R^2^
Step 1					0.020	0.026				0.119	0.125
	Number of medals	147,120.992	420,740.638, 126,498.654	0.443			23,480.042	42,781.999, 89,742.084	0.277		
	Quality performance	100,478.844	−40,228.116, 241,185.803	0.589			3346.853	−30,727.931, 37,421.638	0.077		
Step 2					0.611	0.617				0.323	0.333
	Number of medals	162,261.027	335,748.141, 11,226.087	0.489			3537.168	54,937.113, 62,011.450	0.042		
	Quality performance	77,564.905	−11,150.996, 166,280.807	0.455			2725.839	−27,176.086, 32,627.763	0.063		
	Number of high-level athletes	14,810.446	9476.243, 20,144.650	0.259 **			6918.991	5121.084, 8716.898	0.474 **		
	Number of sport licences	38.133	32,185, 44.081	0.608 **			0.469	−1.535, 2.474	0.029		
	Number of clubs	71.033	−83,403, 225.470	0.038			43.558	−8.495, 95.612	0.092		

Note. ** *p* < 0.01.

**Table 5 ijerph-19-12401-t005:** Multiple regression model of predictors associated with the financing of non-Olympic sport federations.

		Total Budget					CSD Funding				
	Variables	B	95% CI for B	*β*	Adj. R^2^	R^2^	B	95% CI for B	*β*	Adj. R^2^	R^2^
Step 1					0.004	0.009				0.112	0.117
	Number of medals	1,093,249.702	2,411,423.149, 224,923.744	−0.630			18,248.756	3429.463, 6931.952	0.520		
	Quality performance	495,183.764	−15,2669.826, 1,143,037.354	0.581			14,656.443	2280.673, 27,032.213	0.850 *		
Step 2					0.880	0.882				0.243	0.253
	Number of medals	552,665.075	1,020,365.007, −84,965.144	0.319 *			27,854.317	51,659.571, 4049.062	0.794 *		
	Quality performance	157,662.316	−68,844.488, 384,169.121	0.185			16,094.085	4565.215, 27,622.956	0.933 **		
	Number of high-level athletes	54,859.645	36,870.383, 72,848.906	0.145 **			2666.755	1751.128, 3582.383	0.349 **		
	Number of sport licences	−10.313	−22.133, 1.506	−0.077			0.422	−0.180, 1.023	0.156		
	Number of clubs	5719.232	5164.285, 6274.180	0.939 **			−3.788	−32.034, 24.458	−0.031		

Note. * *p* < 0.05, ** *p* < 0.01.

## Data Availability

The data presented in this study are available on request from the corresponding author.

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
