# Peer review of "Understanding the Predictors of Economic Politics on Elite Sport: A Case Study from Spain"

_ijerph, 2022, doi:10.3390/ijerph191912401_

Round 1
Reviewer 1 Report
This paper is well structured and has a sufficient theoretical background. The data source of this paper is clear and comprehensive, and the steps of the research methodology are presented concretely. But there are still some issues that could be further refined.
1. There are some grammatical errors and inaccurate words in the paper, please check and revise them. For example, “t-test compare Olympic and non-Olympic…” and “de facto”. The paper would benefit from thorough proofreading.
2. The literature review is developed step by step, but could you please be more explicit about why this work is necessary and why this methodology is chosen?
3. This study, according to the authors, leads to some similar facts to the bibliography, which further proves the views of the previous works. But compared with previous studies, it would be nice if you show clearly the theoretical contribution of this work.
4. It would be interesting for readers if you could highlight the novelty of practical applications. How these observations in this work would contribute to specific decision-makers?
Author Response
Dear Reviewer,
Please see the attached document.
Thank you very much for your comments. We appreciate your thoughtful and constructive advice. Below, we try to respond to each of the issues raised in your review. In the new version of the paper, we have incorporated changes to address some of these suggestions and we believe that, as a result, the paper has been significantly improved. New text is in red.
Kind regards,
The authors

Reviewer 2 Report
Problem well defined and clear. The state of the art is complete and well structured, showing the variables associated with the success of elite sport, as well as the characterization of the Spanish situation regarding elite sport, comparing the associations that depend on public and private funds and their relationship with the results obtained and how these results are measured.
The study objectives are also well defined and clear.
On page 4 in the first line there is a mistake, which reads “which de facto turns” should read “which in fact turns”.
The description of the method used is very detailed and complete, as is the statistical treatment of the information collected.
The discussion of the results obtained guarantees useful information for all stakeholders related to the sports communities and, above all, state organizations responsible for allocating funds to elite sports. It will also be very useful information for other countries in the same situation and it would be very interesting to repeat the study in other countries where the state is the main funder of elite sports in order to verify whether or not the results obtained are similar to those obtained in Spain.
Author Response
Dear Reviewer,
We have made the changes you suggested.
We appreciate your responsiveness to our article.
Kind regards,
The authors